# Distribution-Independent PAC Learning of Halfspaces with Massart Noise

**Ilias Diakonikolas**
University of Wisconsin-Madison
ilias@cs.wisc.edu

**Themis Gouleakis**
Max Planck Institute for Informatics
tgouleak@mpi-inf.mpg.de

**Christos Tzamos**
University of Wisconsin-Madison
tzamos@wisc.edu

## Abstract

We study the problem of *distribution-independent* PAC learning of halfspaces in the presence of Massart noise. Specifically, we are given a set of labeled examples $(\mathbf{x}, y)$ drawn from a distribution $\mathcal{D}$ on $\mathbb{R}^{d+1}$ such that the marginal distribution on the unlabeled points $\mathbf{x}$ is arbitrary and the labels $y$ are generated by an unknown halfspace corrupted with Massart noise at noise rate $\eta < 1/2$. The goal is to find a hypothesis $h$ that minimizes the misclassification error $\mathbf{Pr}_{(\mathbf{x},y) \sim \mathcal{D}}[h(\mathbf{x}) \neq y]$.

We give a $\mathrm{poly}(d, 1/\epsilon)$ time algorithm for this problem with misclassification error $\eta + \epsilon$. We also provide evidence that improving on the error guarantee of our algorithm might be computationally hard. Prior to our work, no efficient weak (distribution-independent) learner was known in this model, even for the class of disjunctions. The existence of such an algorithm for halfspaces (or even disjunctions) has been posed as an open question in various works, starting with Sloan (1988), Cohen (1997), and was most recently highlighted in Avrim Blum's FOCS 2003 tutorial.

## 1   Introduction

Halfspaces, or Linear Threshold Functions (henceforth LTFs), are Boolean functions $f : \mathbb{R}^d \to \{\pm 1\}$ of the form $f(\mathbf{x}) = \mathrm{sign}(\langle \mathbf{w}, \mathbf{x} \rangle - \theta)$, where $\mathbf{w} \in \mathbb{R}^d$ is the weight vector and $\theta \in \mathbb{R}$ is the threshold. (The function $\mathrm{sign} : \mathbb{R} \to \{\pm 1\}$ is defined as $\mathrm{sign}(u) = 1$ if $u \geq 0$ and $\mathrm{sign}(u) = -1$ otherwise.) The problem of learning an unknown halfspace is as old as the field of machine learning — starting with Rosenblatt's Perceptron algorithm [Ros58] — and has arguably been the most influential problem in the development of the field. In the realizable setting, LTFs are known to be efficiently learnable in Valiant's distribution-independent PAC model [Val84] via Linear Programming [MT94]. In the presence of corrupted data, the situation is more subtle and crucially depends on the underlying noise model. In the agnostic model [Hau92, KSS94] – where an adversary is allowed to arbitrarily corrupt an arbitrary $\eta < 1/2$ fraction of the labels – even weak learning is known to be computationally intractable [GR06, FGKP06, Dan16]. On the other hand, in the presence of Random Classification Noise (RCN) [AL88] – where each label is flipped independently with probability exactly $\eta < 1/2$ – a polynomial time algorithm is known [BFKV96, BFKV97].

In this work, we focus on learning halfspaces with Massart noise [MN06]:

**Definition 1.1** (Massart Noise Model). Let $\mathcal{C}$ be a class of Boolean functions over $X = \mathbb{R}^d$, $\mathcal{D}_{\mathbf{x}}$ be an arbitrary distribution over $X$, and $0 \leq \eta < 1/2$. Let $f$ be an unknown target function in $\mathcal{C}$. A *noisy example oracle*, $\mathrm{EX}^{\mathrm{Mas}}(f, \mathcal{D}_{\mathbf{x}}, \eta)$, works as follows: Each time $\mathrm{EX}^{\mathrm{Mas}}(f, \mathcal{D}_{\mathbf{x}}, \eta)$ is invoked, it

returns a labeled example $(\mathbf{x}, y)$, where $\mathbf{x} \sim \mathcal{D}_{\mathbf{x}}$, $y = f(\mathbf{x})$ with probability $1 - \eta(\mathbf{x})$ and $y = -f(\mathbf{x})$ with probability $\eta(\mathbf{x})$, for an *unknown* parameter $\eta(\mathbf{x}) \le \eta$. Let $\mathcal{D}$ denote the joint distribution on $(\mathbf{x}, y)$ generated by the above oracle. A learning algorithm is given i.i.d. samples from $\mathcal{D}$ and its goal is to output a hypothesis $h$ such that with high probability the error $\mathbf{Pr}_{(\mathbf{x}, y) \sim \mathcal{D}}[h(\mathbf{x}) \ne y]$ is small.

An equivalent formulation of the Massart model [Slo88, Slo92] is the following: With probability $1 - \eta$, we have that $y = f(\mathbf{x})$, and with probability $\eta$ the label $y$ is controlled by an adversary. Hence, the Massart model lies in between the RCN and the agnostic models. (Note that the RCN model corresponds to the special case that $\eta(\mathbf{x}) = \eta$ for all $\mathbf{x} \in X$.) It is well-known (see, e.g., [MN06]) that $\mathrm{poly}(d, 1/\epsilon)$ samples information-theoretically suffice to compute a hypothesis with misclassification error $\mathrm{OPT} + \epsilon$, where $\mathrm{OPT}$ is the misclassification error of the optimal halfspace. Also note that $\mathrm{OPT} \le \eta$ by definition. The question is whether a polynomial time algorithm exists.

The existence of an efficient distribution-independent learning algorithm for halfspaces (or even disjunctions) in the Massart model has been posed as an open question in a number of works. In the first COLT conference [Slo88] (see also [Slo92]), Sloan defined the malicious misclassification noise model (an equivalent formulation of Massart noise, described above) and asked whether there exists an efficient learning algorithm for disjunctions in this model. About a decade later, Cohen [Coh97] asked the same question for the more general class of all LTFs. The question remained open — even for weak learning of disjunctions! — and was highlighted in Avrim Blum's FOCS 2003 tutorial [Blu03]. Specifically, prior to this work, even the following very basic special case remained open:

> *Given labeled examples from an unknown disjunction, corrupted with $1\%$ Massart noise, can we efficiently find a hypothesis that achieves misclassification error $49\%$?*

The reader is referred to slides 39-40 of Avrim Blum's FOCS'03 tutorial [Blu03], where it is suggested that the above problem might be easier than agnostically learning disjunctions. As a corollary of our main result (Theorem 1.2), we answer this question in the affirmative. In particular, we obtain an efficient algorithm that achieves misclassification error arbitrarily close to $\eta$ for all LTFs.

## 1.1 Our Results

The main result of this paper is the following:

**Theorem 1.2** (Main Result). *There is an algorithm that for all $0 < \eta < 1/2$, on input a set of i.i.d. examples from a distribution $\mathcal{D} = \mathrm{EX}^{\mathrm{Mas}}(f, \mathcal{D}_{\mathbf{x}}, \eta)$ on $\mathbb{R}^{d+1}$, where $f$ is an unknown halfspace on $\mathbb{R}^d$, it runs in $\mathrm{poly}(d, b, 1/\epsilon)$ time, where $b$ is an upper bound on the bit complexity of the examples, and outputs a hypothesis $h$ that with high probability satisfies $\mathbf{Pr}_{(\mathbf{x}, y) \sim \mathcal{D}}[h(\mathbf{x}) \ne y] \le \eta + \epsilon$.*

See Theorem 2.9 for a more detailed formal statement. For large-margin halfspaces, we obtain a slightly better error guarantee; see Theorem 2.2 and Remark 2.6.

**Discussion.** We note that our algorithm is non-proper, i.e., the hypothesis $h$ itself is not a halfspace. The polynomial dependence on $b$ in the runtime cannot be removed, even in the noiseless case, unless one obtains strongly-polynomial algorithms for linear programming. Finally, we note that the misclassification error of $\eta$ translates to error $2\eta + \epsilon$ with respect to the target LTF.

Our algorithm gives error $\eta + \epsilon$, instead of the information-theoretic optimum of $\mathrm{OPT} + \epsilon$. To complement our positive result, we provide some evidence that improving on our $(\eta + \epsilon)$ error guarantee may be challenging. Roughly speaking, we show (see Theorems B.1 and B.2 in the supplementary material) that natural approaches — involving convex surrogates and refinements thereof — inherently fail, even under margin assumptions. (See Section 1.2 for a discussion.)

**Broader Context.** This work is part of the broader agenda of designing robust estimators in the distribution-independent setting with respect to natural noise models. A recent line of work [KLS09, ABL17, DKK+16, LRV16, DKK+17, DKK+18, DKS18, KKM18, DKS19, DKK+19] has given efficient robust estimators for a range of learning tasks (both supervised and unsupervised) in the presence of a small constant fraction of adversarial corruptions. A limitation of these results is the assumption that the good data comes from a "tame" distribution, e.g., Gaussian or isotropic log-concave distribution. On the other hand, if *no* assumption is made on the good data and the noise remains fully adversarial, these problems become computationally intractable [Ber06, GR06, Dan16].

This suggests the following general question: *Are there realistic noise models that allow for efficient algorithms without imposing (strong) assumptions on the good data?* Conceptually, the algorithmic results of this paper could be viewed as an affirmative answer to this question for the problem of learning halfspaces.

## 1.2 Technical Overview

In this section, we provide an outline of our approach and a comparison to previous techniques. Since the distribution on the unlabeled data is arbitrary, we can assume w.l.o.g. that the threshold $\theta = 0$.

**Massart Noise versus RCN.** Random Classification Noise (RCN) [AL88] is the special case of Massart noise where each label is flipped with probability *exactly* $\eta < 1/2$. At first glance, it might seem that Massart noise is easier to deal with computationally than RCN. After all, in the Massart model we add *at most as much noise* as in the RCN model. It turns out that this intuition is fundamentally flawed. Roughly speaking, the ability of the Massart adversary to choose *whether* to perturb a given label and, if so, with what probability (which is *unknown* to the learner), makes the design of efficient algorithms in this model challenging. In particular, the well-known connection between learning with RCN and the Statistical Query (SQ) model [Kea93, Kea98] no longer holds, i.e., the property of being an SQ algorithm does *not* automatically suffice for noise-tolerant learning with Massart noise. We note that this connection with the SQ model is leveraged in [BFKV96, BFKV97] to obtain their polynomial time algorithm for learning halfspaces with RCN.

**Large Margin Halfspaces.** To illustrate our approach, we start by describing our learning algorithm for $\gamma$-*margin* halfspaces on the unit ball. That is, we assume $|\langle \mathbf{w}^*, \mathbf{x} \rangle| \geq \gamma$ for every $\mathbf{x}$ in the support, where $\mathbf{w}^* \in \mathbb{R}^d$ with $\|\mathbf{w}^*\|_2 = 1$ defines the target halfspace $h_{\mathbf{w}^*}(\mathbf{x}) = \mathrm{sign}(\langle \mathbf{w}^*, \mathbf{x} \rangle)$. Our goal is to design a $\mathrm{poly}(d, 1/\epsilon, 1/\gamma)$ time learning algorithm in the presence of Massart noise.

In the RCN model, the large margin case is easy because the learning problem is essentially convex. That is, there is a convex surrogate that allows us to formulate the problem as a convex program. We can use SGD to find a near-optimal solution to this convex program, which automatically gives a *strong proper* learner. This simple fact does not appear explicitly in the literature, but follows easily from standard tools. [Byl94] showed that a variant of the Perceptron algorithm (which can be viewed as gradient descent on a particular convex objective) learns $\gamma$-margin halfspaces in $\mathrm{poly}(d, 1/\epsilon, 1/\gamma)$ time. The algorithm in [Byl94] requires an additional anti-concentration condition about the distribution, which is easy to remove. In Appendix C, we show that a "smoothed" version of Bylander's objective suffices as a convex surrogate under only the margin assumption.

Roughly speaking, the reason that a convex surrogate works for RCN is that the expected effect of the noise on each label is known a priori. Unfortunately, this is not the case for Massart noise. We show (Theorem B.1 in Appendix B) that no convex surrogate can lead to a *weak learner*, even under a margin assumption. That is, if $\widehat{\mathbf{w}}$ is the minimizer of $G(\mathbf{w}) = \mathbf{E}_{(\mathbf{x},y) \sim \mathcal{D}}[\phi(y \langle \mathbf{w}, \mathbf{x} \rangle)]$, where $\phi$ can be any convex function, then the hypothesis $\mathrm{sign}(\langle \widehat{\mathbf{w}}, \mathbf{x} \rangle)$ is not even a weak learner. So, in sharp contrast with the RCN case, the problem is non-convex in this sense.

Our Massart learning algorithm for large margin halfspaces still uses a convex surrogate, but in a qualitatively different way. Instead of attempting to solve the problem in one-shot, our algorithm adaptively applies a sequence of convex optimization problems to obtain an accurate solution in disjoint subsets of the space. Our iterative approach is motivated by a new structural lemma (Lemma 2.5) establishing the following: *Even though minimizing a convex proxy does not lead to small misclassification error over the entire space, there exists a region with non-trivial probability mass where it does.* Moreover, this region is efficiently identifiable by a simple thresholding rule. Specifically, we show that there exists a threshold $T > 0$ (which can be found algorithmically) such that the hypothesis $\mathrm{sign}(\langle \widehat{\mathbf{w}}, \mathbf{x} \rangle)$ has error bounded by $\eta + \epsilon$ in the region $R_T = \{\mathbf{x} : |\langle \widehat{\mathbf{w}}, \mathbf{x} \rangle| \geq T\}$. Here $\widehat{\mathbf{w}}$ is any near-optimal solution to an appropriate convex optimization problem, defined via a convex surrogate objective similar to the one used in [Byl94]. We note that Lemma 2.5 is the main technical novelty of this paper and motivates our algorithm. Given Lemma 2.5, in any iteration $i$ we can find the best threshold $T^{(i)}$ using samples, and obtain a learner with misclassification error $\eta + \epsilon$ in the corresponding region. Since each region has non-trivial mass, iterating this scheme a small number of times allows us to find a non-proper hypothesis (a decision-list of halfspaces) with misclassification error at most $\eta + \epsilon$ in the entire space.

The idea of iteratively optimizing a convex surrogate was used in [BFKV96] to learn halfspaces with RCN *without* a margin. Despite this similarity, we note that the algorithm of [BFKV96] fails to even obtain a weak learner in the Massart model. We point out two crucial technical differences: First, the iterative approach in [BFKV96] was needed to achieve polynomial running time. As mentioned already, a convex proxy is guaranteed to converge to the true solution with RCN, but the convergence may be too slow (when the margin is tiny). In contrast, with Massart noise (even under a margin condition) convex surrogates cannot even give weak learning in the entire domain. Second, the algorithm of [BFKV96] used a fixed threshold in each iteration, equal to the margin parameter obtained after an appropriate pre-processing of the data (that is needed in order to ensure a weak margin property). In contrast, in our setting, we need to find an appropriate threshold $T^{(i)}$ in each iteration $i$, according to the criterion specified by our Lemma 2.5.

**General Case.** Our algorithm for the general case (in the absence of a margin) is qualitatively similar to our algorithm for the large margin case, but the details are more elaborate. We borrow an idea from [BFKV96] that in some sense allows us to "reduce" the general case to the large margin case. Specifically, [BFKV96] (see also [DV04a]) developed a pre-processing routine that slightly modifies the distribution on the unlabeled points and guarantees the following *weak margin* property: After pre-processing, there exists an explicit margin parameter $\sigma = \Omega(1/\text{poly}(d, b))$, such that any hyperplane through the origin has at least a non-trivial mass of the distribution at distance at least $\sigma$ from it. Using this pre-processing step, we are able to adapt our algorithm from the previous subsection to work without margin assumptions in $\text{poly}(d, b, 1/\epsilon)$ time. While our analysis is similar in spirit to the case of large margin, we note that the margin property obtained via the [BFKV96, DV04a] preprocessing step is (necessarily) weaker, hence additional careful analysis is required.

**Lower Bounds Against Natural Approaches.** We have already explained our Theorem B.1, which shows that using a convex surrogate over the entire space cannot not give a weak learner. Our algorithm, however, can achieve error $\eta + \epsilon$ by iteratively optimizing a specific convex surrogate in disjoint subsets of the domain. A natural question is whether one can obtain qualitatively better accuracy, e.g., $f(\text{OPT}) + \epsilon$, by using a *different* convex objective function in our iterative thresholding approach. We show (Theorem B.2) that such an improvement is not possible: Using a different convex proxy cannot lead to error better than $(1 - o(1)) \cdot \eta$. It is a plausible conjecture that improving on the error guarantee of our algorithm is computationally hard. We leave this as an intriguing open problem for future work.

## 1.3 Prior and Related Work

Bylander [Byl94] gave a polynomial time algorithm to learn large margin halfspaces with RCN (under an additional anti-concentration assumption). The work of Blum *et al.* [BFKV96, BFKV97] gave the first polynomial time algorithm for distribution-independent learning of halfspaces with RCN without any margin assumptions. Soon thereafter, [Coh97] gave a polynomial-time proper learning algorithm for the problem. Subsequently, Dunagan and Vempala [DV04b] gave a rescaled perceptron algorithm for solving linear programs, which translates to a significantly simpler and faster proper learning algorithm.

The term "Massart noise" was coined after [MN06]. An equivalent version of the model was previously studied by Rivest and Sloan [Slo88, Slo92, RS94, Slo96], and a very similar asymmetric random noise model goes back to Vapnik [Vap82]. Prior to this work, essentially no efficient algorithms with non-trivial error guarantees were known in the distribution-free Massart noise model. It should be noted that polynomial time algorithms with error $\text{OPT} + \epsilon$ are known [ABHU15, ZLC17, YZ17] when the marginal distribution on the unlabeled data is uniform on the unit sphere. For the case that the unlabeled data comes from an isotropic log-concave distribution, [ABHZ16] give a $d^{2^{\text{poly}(1/(1-2\eta))}}/\text{poly}(\epsilon)$ sample and time algorithm.

## 1.4 Preliminaries

For $n \in \mathbb{Z}_+$, we denote $[n] \stackrel{\text{def}}{=} \{1, \ldots, n\}$. We will use small boldface characters for vectors and we let $\mathbf{e}_i$ denote the $i$-th vector of an orthonormal basis.

For $\mathbf{x} \in \mathbb{R}^d$, and $i \in [d]$, $\mathbf{x}_i$ denotes the $i$-th coordinate of $\mathbf{x}$, and $\|\mathbf{x}\|_2 \overset{\text{def}}{=} (\sum_{i=1}^d \mathbf{x}_i^2)^{1/2}$ denotes the $\ell_2$-norm of $\mathbf{x}$. We will use $\langle \mathbf{x}, \mathbf{y} \rangle$ for the inner product between $\mathbf{x}, \mathbf{y} \in \mathbb{R}^d$. We will use $\mathbf{E}[X]$ for the expectation of random variable $X$ and $\mathbf{Pr}[\mathcal{E}]$ for the probability of event $\mathcal{E}$.

An origin-centered halfspace is a Boolean-valued function $h_{\mathbf{w}} : \mathbb{R}^d \to \{\pm 1\}$ of the form $h_{\mathbf{w}}(\mathbf{x}) = \text{sign}(\langle \mathbf{w}, \mathbf{x} \rangle)$, where $\mathbf{w} \in \mathbb{R}^d$. (Note that we may assume w.l.o.g. that $\|\mathbf{w}\|_2 = 1$.) We denote by $\mathcal{H}_d$ the class of all origin-centered halfspaces on $\mathbb{R}^d$.

We consider a classification problem where labeled examples $(\mathbf{x}, y)$ are drawn i.i.d. from a distribution $\mathcal{D}$. We denote by $\mathcal{D}_{\mathbf{x}}$ the marginal of $\mathcal{D}$ on $\mathbf{x}$, and for any $\mathbf{x}$ denote $\mathcal{D}_y(\mathbf{x})$ the distribution of $y$ conditional on $\mathbf{x}$. Our goal is to find a hypothesis classifier $h$ with low misclassification error. We will denote the misclassification error of a hypothesis $h$ with respect to $\mathcal{D}$ by $\text{err}_{0-1}^{\mathcal{D}}(h) = \mathbf{Pr}_{(\mathbf{x}, y) \sim \mathcal{D}}[h(\mathbf{x}) \neq y]$. Let $\text{OPT} = \min_{h \in \mathcal{H}_d} \text{err}_{0-1}^{\mathcal{D}}(h)$ denote the optimal misclassification error of any halfspace, and $\mathbf{w}^*$ be the normal vector to a halfspace $h_{\mathbf{w}^*}$ that achieves this.

## 2 Algorithm for Learning Halfspaces with Massart Noise

In this section, we present the main result of this paper, which is an efficient algorithm that achieves $\eta + \epsilon$ misclassification error for distribution-independent learning of halfspaces with Massart noise $\eta$.

Our algorithm uses (stochastic) gradient descent on a convex proxy function $L(\mathbf{w})$ for the misclassification error to identify a region with small misclassification error. The loss function penalizes the points which are misclassified by the threshold function $h_{\mathbf{w}}$, proportionally to the distance from the corresponding hyperplane, while rewards the correctly classified points at a smaller rate. Directly optimizing this convex objective does not lead to a separator with low error, but guarantees that for a non-negligible fraction of the mass away from the separating hyperplane the misclassification error will be at most $\eta + \epsilon$. Classifying points in this region according to the hyperplane and recursively working on the remaining points, we obtain an improper learning algorithm that achieves $\eta + \epsilon$ error overall.

We now develop some necessary notation before proceeding with the description and analysis of our algorithm.

Our algorithm considers the following convex proxy for the misclassification error as a function of the weight vector $\mathbf{w}$:

$$L(\mathbf{w}) = \underset{(\mathbf{x}, y) \sim \mathcal{D}}{\mathbf{E}}[\text{LeakyRelu}_\lambda(-y \langle \mathbf{w}, \mathbf{x} \rangle)] \,,$$

under the constraint $\|\mathbf{w}\|_2 \leq 1$, where $\text{LeakyRelu}_\lambda(z) = \begin{cases} (1 - \lambda)z & \text{if } z \geq 0 \\ \lambda z & \text{if } z < 0 \end{cases}$ and $\lambda$ is the *leakage* parameter, which we will set to be $\lambda \approx \eta$.

We define the per-point misclassification error and the error of the proxy function as $\text{err}(\mathbf{w}, \mathbf{x}) = \mathbf{Pr}_{y \sim \mathcal{D}_y(\mathbf{x})}[\mathbf{w}(\mathbf{x}) \neq y]$ and $\ell(\mathbf{w}, \mathbf{x}) = \mathbf{E}_{y \sim \mathcal{D}_y(\mathbf{x})}[\text{LeakyRelu}_\lambda(-y \langle \mathbf{w}, \mathbf{x} \rangle)]$ respectively.

Notice that $\text{err}_{0-1}^{\mathcal{D}}(h_{\mathbf{w}}) = \mathbf{E}_{\mathbf{x} \sim \mathcal{D}_{\mathbf{x}}}[\text{err}(\mathbf{w}, \mathbf{x})]$ and $L(\mathbf{w}) = \mathbf{E}_{\mathbf{x} \sim \mathcal{D}_{\mathbf{x}}}[\ell(\mathbf{w}, \mathbf{x})]$. Moreover, $\text{OPT} = \mathbf{E}_{\mathbf{x} \sim \mathcal{D}_{\mathbf{x}}}[\text{err}(\mathbf{w}^*, \mathbf{x})] = \mathbf{E}_{\mathbf{x} \sim \mathcal{D}_{\mathbf{x}}}[\eta(\mathbf{x})]$.

**Relationship between proxy loss and misclassification error** We first relate the proxy loss and the misclassification error.

**Claim 2.1.** *For any $\mathbf{w}, \mathbf{x}$, we have that $\ell(\mathbf{w}, \mathbf{x}) = (\text{err}(\mathbf{w}, \mathbf{x}) - \lambda)|\langle \mathbf{w}, \mathbf{x} \rangle|$.*

*Proof.* We consider two cases:

- **Case** $\text{sign}(\langle \mathbf{w}, \mathbf{x} \rangle) = \text{sign}(\langle \mathbf{w}^*, \mathbf{x} \rangle)$: In this case, we have that $\text{err}(\mathbf{w}, \mathbf{x}) = \eta(\mathbf{x})$, while $\ell(\mathbf{w}, \mathbf{x}) = \eta(\mathbf{x})(1 - \lambda)|\langle \mathbf{w}, \mathbf{x} \rangle| - (1 - \eta(\mathbf{x}))\lambda|\langle \mathbf{w}, \mathbf{x} \rangle| = (\eta(\mathbf{x}) - \lambda)|\langle \mathbf{w}, \mathbf{x} \rangle|$.

- **Case** $\text{sign}(\langle \mathbf{w}, \mathbf{x} \rangle) \neq \text{sign}(\langle \mathbf{w}^*, \mathbf{x} \rangle)$: In this case, we have that $\text{err}(\mathbf{w}, \mathbf{x}) = 1 - \eta(\mathbf{x})$, while $\ell(\mathbf{w}, \mathbf{x}) = (1 - \eta(\mathbf{x}))(1 - \lambda)|\langle \mathbf{w}, \mathbf{x} \rangle| - \eta(\mathbf{x})\lambda|\langle \mathbf{w}, \mathbf{x} \rangle| = (1 - \eta(\mathbf{x}) - \lambda)|\langle \mathbf{w}, \mathbf{x} \rangle|$.

This completes the proof of Claim 2.1. □

Claim 2.1 shows that minimizing $\mathbf{E}_{\mathbf{x} \sim \mathcal{D}_{\mathbf{x}}} \left[ \frac{\ell(\mathbf{w}, \mathbf{x})}{|\langle \mathbf{w}, \mathbf{x} \rangle|} \right]$ is equivalent to minimizing the misclassification error. Unfortunately, this objective is hard to minimize as it is non-convex, but one would hope that minimizing $L(\mathbf{w})$ instead may have a similar effect. As we show, this is not true because $|\langle \mathbf{w}, \mathbf{x} \rangle|$ might vary significantly across points, and in fact it is not possible to use a convex proxy that achieves bounded misclassification error directly.

Our algorithm circumvents this difficulty by approaching the problem indirectly to find a non-proper classifier. Specifically, our algorithm works in multiple rounds, where within each round only points with high value of $|\langle \mathbf{w}, \mathbf{x} \rangle|$ are considered. The intuition is based on the fact that the approximation of the convex proxy to the misclassification error is more accurate for those points that have comparable distance to the halfspace.

In Section 2.1, we handle the large margin case and in Section 2.2 we handle the general case.

## 2.1 Warm-up: Learning Large Margin Halfspaces

We consider the case that there is no probability mass within distance $\gamma$ from the separating hyperplane $\langle \mathbf{w}^*, \mathbf{x} \rangle = 0$, $\|\mathbf{w}^*\|_2 = 1$. Formally, assume that for every $\mathbf{x} \sim \mathcal{D}_{\mathbf{x}}$, $\|\mathbf{x}\|_2 \leq 1$ and that $\langle \mathbf{w}^*, \mathbf{x} \rangle \geq \gamma$.

The pseudo-code of our algorithm is given in Algorithm 1. Our algorithm returns a decision list $[(\mathbf{w}^{(1)}, T^{(1)}), (\mathbf{w}^{(2)}, T^{(2)}), \cdots]$ as output. To classify a point $\mathbf{x}$ given the decision list, the first $i$ is identified such that $|\langle \mathbf{w}^{(i)}, \mathbf{x} \rangle| \geq T^{(i)}$ and $\mathrm{sign}(\langle \mathbf{w}^{(i)}, \mathbf{x} \rangle)$ is returned. If no such $i$ exists, an arbitrary prediction is returned.

---

**Algorithm 1** Main Algorithm (with margin)

---

1: Set $S^{(1)} = \mathbb{R}^d$, $\lambda = \eta + \epsilon$, $m = \tilde{O}(\frac{1}{\gamma^2 \epsilon^4})$.
2: Set $i \leftarrow 1$.
3: Draw $O\left((1/\epsilon^2) \log(1/(\epsilon\gamma))\right)$ samples from $\mathcal{D}_{\mathbf{x}}$ to form an empirical distribution $\tilde{\mathcal{D}}_{\mathbf{x}}$.
4: **while** $\mathbf{Pr}_{\mathbf{x} \sim \tilde{\mathcal{D}}_{\mathbf{x}}} \left[ \mathbf{x} \in S^{(i)} \right] \geq \epsilon$ **do**
5:     Set $\mathcal{D}^{(i)} = \mathcal{D}|_{S^{(i)}}$, the distribution conditional on the unclassified points.
6:     Let $L^{(i)}(\mathbf{w}) = \mathbf{E}_{(\mathbf{x}, y) \sim \mathcal{D}^{(i)}}[\mathrm{LeakyRelu}_\lambda(-y\langle \mathbf{w}, \mathbf{x} \rangle)]$
7:     Run SGD on $L^{(i)}(\mathbf{w})$ for $\tilde{O}(1/(\gamma^2 \epsilon^2))$ iterations to get $\mathbf{w}^{(i)}$ with $\|\mathbf{w}^{(i)}\|_2 = 1$ such that $L^{(i)}(\mathbf{w}^{(i)}) \leq \min_{\mathbf{w}: \|\mathbf{w}\|_2 \leq 1} L^{(i)}(\mathbf{w}) + \gamma\epsilon/2$.
8:     Draw $m$ samples from $\mathcal{D}^{(i)}$ to form an empirical distribution $\mathcal{D}_m^{(i)}$.
9:     Find a threshold $T^{(i)}$ such that $\mathbf{Pr}_{(\mathbf{x}, y) \sim \mathcal{D}_m^{(i)}}[|\langle \mathbf{w}^{(i)}, \mathbf{x} \rangle| \geq T^{(i)}] \geq \gamma\epsilon$ and the empirical misclassification error, $\mathbf{Pr}_{(\mathbf{x}, y) \sim \mathcal{D}_m^{(i)}}[h_{\mathbf{w}^{(i)}}(\mathbf{x}) \neq y \mid |\langle \mathbf{w}^{(i)}, \mathbf{x} \rangle| \geq T^{(i)}]$, is minimized.
10:     Update the unclassified region $S^{(i+1)} \leftarrow S^{(i)} \setminus \{\mathbf{x} : |\langle \mathbf{w}^{(i)}, \mathbf{x} \rangle| \geq T^{(i)}\}$ and set $i \leftarrow i + 1$.
11: Return the classifier $[(\mathbf{w}^{(1)}, T^{(1)}), (\mathbf{w}^{(2)}, T^{(2)}), \cdots]$

---

The main result of this section is the following:

**Theorem 2.2.** *Let $\mathcal{D}$ be a distribution on $\mathbb{B}_d \times \{\pm 1\}$ such that $\mathcal{D}_{\mathbf{x}}$ satisfies the $\gamma$-margin property with respect to $\mathbf{w}^*$ and $y$ is generated by $\mathrm{sign}(\langle \mathbf{w}^*, \mathbf{x} \rangle)$ corrupted with Massart noise at rate $\eta < 1/2$. Algorithm 1 uses $\tilde{O}(1/(\gamma^3 \epsilon^5))$ samples from $\mathcal{D}$, runs in $\mathrm{poly}(d, 1/\epsilon, 1/\gamma)$ time, and returns, with probability $2/3$, a classifier $h$ with misclassification error $\mathrm{err}_{0-1}^{\mathcal{D}}(h) \leq \eta + \epsilon$.*

Our analysis focuses on a single iteration of Algorithm 1. We will show that a large fraction of the points is classified at every iteration within error $\eta + \epsilon$. To achieve this, we analyze the convex objective $L$. We start by showing that the optimal classifier $\mathbf{w}^*$ obtains a significantly negative objective value.

**Lemma 2.3.** *If $\lambda \geq \eta$, then $L(\mathbf{w}^*) \leq -\gamma(\lambda - \mathrm{OPT})$.*

*Proof.* For any fixed $\mathbf{x}$, using Claim 2.1, we have that

$$\ell(\mathbf{w}^*, \mathbf{x}) = (\mathrm{err}(\mathbf{w}^*, \mathbf{x}) - \lambda)|\langle \mathbf{w}^*, \mathbf{x} \rangle| = (\eta(\mathbf{x}) - \lambda)|\langle \mathbf{w}^*, \mathbf{x} \rangle| \leq -\gamma(\lambda - \eta(\mathbf{x})) ,$$

since $|\langle \mathbf{w}^*, \mathbf{x} \rangle| \geq \gamma$ and $\eta(\mathbf{x}) - \lambda \leq 0$. Taking expectation over $\mathbf{x} \sim \mathcal{D}_{\mathbf{x}}$, the statement follows. $\qquad \square$

Lemma 2.3 is the only place where the Massart noise assumption is used in our approach and establishes that points with sufficiently negative value exist. As we will show, any weight vector $\mathbf{w}$ with this property can be found with few samples and must accurately classify some region of non-negligible mass away from it (Lemma 2.5).

We now argue that we can use stochastic gradient descent (SGD) to efficiently identify a point $\mathbf{w}$ that achieves comparably small objective value to the guarantee of Lemma 2.3. We use the following standard property of SGD:

**Lemma 2.4** (see, e.g., Theorem 3.4.11 in [Duc16]). *Let $L$ be any convex function. Consider the (projected) SGD iteration that is initialized at $\mathbf{w}^{(0)} = \mathbf{0}$ and for every step computes*

$$\mathbf{w}^{(t+\frac{1}{2})} = \mathbf{w}^{(t)} - \rho \mathbf{v}^{(t)} \quad and \quad \mathbf{w}^{(t+1)} = \arg\min_{\mathbf{w}:\|\mathbf{w}\|_2 \leq 1} \left\| \mathbf{w} - \mathbf{w}^{(t+\frac{1}{2})} \right\|_2 ,$$

*where $\mathbf{v}^{(t)}$ is a stochastic gradient such that for all steps $\mathbf{E}[\mathbf{v}^{(t)}|\mathbf{w}^{(t)}] \in \partial L(\mathbf{w}^{(t)})$ and $\left\| \mathbf{v}^{(t)} \right\|_2 \leq 1$. Assume that SGD is run for $T$ iterations with step size $\rho = \frac{1}{\sqrt{T}}$ and let $\bar{w} = \frac{1}{T} \sum_{t=1}^{T} \mathbf{w}^{(t)}$. Then, for any $\epsilon, \delta > 0$, after $T = \Omega(\log(1/\delta)/\epsilon^2)$ iterations with probability with probability at least $1 - \delta$ we have that $L(\bar{\mathbf{w}}) \leq \min_{\mathbf{w}:\|\mathbf{w}\|_2 \leq 1} L(\mathbf{w}) + \epsilon$.*

By Lemma 2.3, we know that $\min_{\mathbf{w}:\|\mathbf{w}\|_2 \leq 1} L(\mathbf{w}) \leq -\gamma(\lambda - \mathrm{OPT})$. By Lemma 2.4, it follows that by running SGD on $L(\mathbf{w})$ with projection to the unit $\ell_2$-ball for $O\left(\log(1/\delta)/(\gamma^2(\lambda - \mathrm{OPT})^2)\right)$ steps, we find a $\mathbf{w}$ such that $L(\mathbf{w}) \leq -\gamma(\lambda - \mathrm{OPT})/2$ with probability at least $1 - \delta$.

Note that we can assume without loss of generality that $\|\mathbf{w}\|_2 = 1$, as increasing the magnitude of $\mathbf{w}$ only decreases the objective value.

We now consider the misclassification error of the halfspace $h_\mathbf{w}$ conditional on the points that are further than some distance $T$ from the separating hyperplane. We claim that there exists a threshold $T > 0$ where the restriction has non-trivial mass and the conditional misclassification error is small:

**Lemma 2.5.** *Consider a vector $\mathbf{w}$ with $L(\mathbf{w}) < 0$. There exists a threshold $T \geq 0$ such that (i) $\mathbf{Pr}_{(\mathbf{x},y)\sim\mathcal{D}}[|\langle \mathbf{w}, \mathbf{x}\rangle| \geq T] \geq \frac{|L(\mathbf{w})|}{2\lambda}$, and (ii) $\mathbf{Pr}_{(\mathbf{x},y)\sim\mathcal{D}}[h_\mathbf{w}(\mathbf{x}) \neq y \mid |\langle \mathbf{w}, \mathbf{x}\rangle| \geq T] \leq \lambda - \frac{|L(\mathbf{w})|}{2}$.*

*Proof.* We will show there is a $T \geq 0$ such that $\mathbf{Pr}_{(\mathbf{x},y)\sim\mathcal{D}}[h_\mathbf{w}(\mathbf{x}) \neq y \mid |\langle \mathbf{w}, \mathbf{x}\rangle| \geq T] \leq \lambda - \zeta$, where $\zeta \stackrel{\text{def}}{=} |L(\mathbf{w})|/2$, or equivalently, $\mathbf{E}_{\mathbf{x}\sim\mathcal{D}_\mathbf{x}}[(\mathrm{err}(\mathbf{w},\mathbf{x}) - \lambda + \zeta)\mathbb{1}_{|\langle \mathbf{w},\mathbf{x}\rangle| \geq T}] \leq 0$.

For a $T$ drawn uniformly at random in $[0, 1]$, we have that:

$$\int_0^1 \mathbf{E}_{\mathbf{x}\sim\mathcal{D}_\mathbf{x}}[(\mathrm{err}(\mathbf{w},\mathbf{x}) - \lambda + \zeta)\mathbb{1}_{|\langle \mathbf{w},\mathbf{x}\rangle| \geq T}]dT = \mathbf{E}_{\mathbf{x}\sim\mathcal{D}_\mathbf{x}}[(\mathrm{err}(\mathbf{w},\mathbf{x}) - \lambda)|\langle \mathbf{w}, \mathbf{x}\rangle|] + \zeta\mathbf{E}_{\mathbf{x}\sim\mathcal{D}_\mathbf{x}}[|\langle \mathbf{w}, \mathbf{x}\rangle|]$$

$$\leq \mathbf{E}_{\mathbf{x}\sim\mathcal{D}_\mathbf{x}}[\ell(\mathbf{w},\mathbf{x})] + \zeta = L(\mathbf{w}) + \zeta = L(\mathbf{w})/2 < 0 .$$

Thus, there exists a $\bar{T}$ such that $\mathbf{E}_{\mathbf{x}\sim\mathcal{D}_\mathbf{x}}[(\mathrm{err}(\mathbf{w},\mathbf{x}) - \lambda + \zeta)\mathbb{1}_{|\langle \mathbf{w},\mathbf{x}\rangle| \geq \bar{T}}] \leq 0$. Consider the minimum such $\bar{T}$. Then we have

$$\int_{\bar{T}}^1 \mathbf{E}_{\mathbf{x}\sim\mathcal{D}_\mathbf{x}}[(\mathrm{err}(\mathbf{w},\mathbf{x}) - \lambda + \zeta)\mathbb{1}_{|\langle \mathbf{w},\mathbf{x}\rangle| \geq T}]dT \geq -\lambda \cdot \mathbf{Pr}_{(\mathbf{x},y)\sim\mathcal{D}}[|\langle \mathbf{w}, \mathbf{x}\rangle| \geq \bar{T}] .$$

By definition of $\bar{T}$, it must be the case that $\int_0^{\bar{T}} \mathbf{E}_{\mathbf{x}\sim\mathcal{D}_\mathbf{x}}[(\mathrm{err}(\mathbf{w},\mathbf{x}) - \lambda + \zeta)\mathbb{1}_{|\langle \mathbf{w},\mathbf{x}\rangle| \geq T}]dT \geq 0$. Therefore,

$$\frac{L(\mathbf{w})}{2} \geq \int_{\bar{T}}^1 \mathbf{E}_{\mathbf{x}\sim\mathcal{D}_\mathbf{x}}[(\mathrm{err}(\mathbf{w},\mathbf{x}) - \lambda + \zeta)\mathbb{1}_{|\langle \mathbf{w},\mathbf{x}\rangle| \geq T}]dT \geq -\lambda \cdot \mathbf{Pr}_{(\mathbf{x},y)\sim\mathcal{D}}[|\langle \mathbf{w}, \mathbf{x}\rangle| \geq \bar{T}] ,$$

which implies that $\mathbf{Pr}_{(\mathbf{x},y)\sim\mathcal{D}}[|\langle \mathbf{w}, \mathbf{x}\rangle| \geq \bar{T}] \geq \frac{|L(\mathbf{w})|}{2\lambda}$. This completes the proof of Lemma 2.5. $\qquad\square$

Even though minimizing the convex proxy $L$ does not lead to low misclassification error overall, Lemma 2.5 shows that there exists a region of non-trivial mass where it does. This region is identifiable by a simple threshold rule. We are now ready to prove Theorem 2.2.

*Proof of Theorem 2.2.* We consider the steps of Algorithm 1 in each iteration of the while loop. At iteration $i$, we consider a distribution $\mathcal{D}^{(i)}$ consisting only of points not handled in previous iterations.

We start by noting that with high probability the total number of iterations is $\tilde{O}(1/(\gamma\epsilon))$. This can be seen as follows: The empirical probability mass under $\mathcal{D}_m^{(i)}$ of the region $\{\mathbf{x} : |\langle \mathbf{w}^{(i)}, \mathbf{x}\rangle| \geq T^{(i)}\}$ removed from $S^{(i)}$ to obtain $S^{(i+1)}$ is at least $\gamma\epsilon$ (Step 9). Since $m = \tilde{O}(1/(\gamma^2\epsilon^4))$, the DKW inequality [DKW56] implies that the true probability mass of this region is at least $\gamma\epsilon/2$ with high probability. By a union bound over $i \leq K = \Theta(\log(1/\epsilon)/(\epsilon\gamma))$, it follows that with high probability we have that $\Pr_{\mathcal{D}_\mathbf{x}}[S^{(i+1)}] \leq (1 - \gamma\epsilon/2)^i$ for all $i \in [K]$. After $K$ iterations, we will have that $\Pr_{\mathcal{D}_\mathbf{x}}[S^{(i+1)}] \leq \epsilon/3$. Step 3 guarantees that the mass of $S^{(i)}$ under $\tilde{\mathcal{D}}_\mathbf{x}$ is within an additive $\epsilon/3$ of its mass under $\mathcal{D}_\mathbf{x}$, for $i \in [K]$. This implies that the loop terminates after at most $K$ iterations.

By Lemma 2.3 and the fact that every $\mathcal{D}^{(i)}$ has margin $\gamma$, it follows that the minimizer of the loss $L^{(i)}$ has value less than $-\gamma(\lambda - \mathrm{OPT}^{(i)}) \leq -\gamma\epsilon$, as $\mathrm{OPT}^{(i)} \leq \eta$ and $\lambda = \eta + \epsilon$. By the guarantees of Lemma 2.4, running SGD in line 7 on $L^{(i)}(\cdot)$ with projection to the unit $\ell_2$-ball for $O\left(\log(1/\delta)/(\gamma^2\epsilon^2)\right)$ steps, we obtain a $\mathbf{w}^{(i)}$ such that, with probability at least $1 - \delta$, it holds $L^{(i)}(\mathbf{w}^{(i)}) \leq -\gamma\epsilon/2$ and $\|\mathbf{w}^{(i)}\|_2 = 1$. Here $\delta > 0$ is a parameter that is selected so that the following claim holds: With probability at least $9/10$, for all iterations $i$ of the while loop we have that $L^{(i)}(\mathbf{w}^{(i)}) \leq -\gamma\epsilon/2$. Since the total number of iterations is $\tilde{O}(1/(\gamma\epsilon))$, setting $\delta$ to $\tilde{\Omega}(\epsilon\gamma)$ and applying a union bound over all iterations gives the previous claim. Therefore, the total number of SGD steps per iteration is $\tilde{O}(1/(\gamma^2\epsilon^2))$. For a given iteration of the while loop, running SGD requires $\tilde{O}(1/(\gamma^2\epsilon^2))$ samples from $\mathcal{D}^{(i)}$ which translate to at most $\tilde{O}\left(1/(\gamma^2\epsilon^3)\right)$ samples from $\mathcal{D}$, as $\Pr_{\mathbf{x}\sim\mathcal{D}_\mathbf{x}}\left[\mathbf{x} \in S^{(i)}\right] \geq 2\epsilon/3$.

Lemma 2.5 implies that there exists $T \geq 0$ such that: (a) $\Pr_{(\mathbf{x},y)\sim\mathcal{D}^{(i)}}[|\langle \mathbf{w}, \mathbf{x}\rangle| \geq T] \geq \gamma\epsilon$, and (b) $\Pr_{(\mathbf{x},y)\sim\mathcal{D}^{(i)}}[h_\mathbf{w}(\mathbf{x}) \neq y \mid |\langle \mathbf{w}, \mathbf{x}\rangle| \geq T] \leq \eta + \epsilon$. Line 9 of Algorithm 1 estimates the threshold using samples. By the DKW inequality [DKW56], we know that with $m = \tilde{O}(1/(\gamma^2\epsilon^4))$ samples we can estimate the CDF within error $\gamma\epsilon^2$ with probability $1 - \mathrm{poly}(\epsilon, \gamma)$. This suffices to estimate the probability mass of the region within additive $\gamma\epsilon^2$ and the misclassification error within $\epsilon/3$. This is satisfied for all iterations with constant probability.

In summary, with high constant success probability, Algorithm 1 runs for $\tilde{O}(1/(\gamma\epsilon))$ iterations and draws $\tilde{O}(1/(\gamma^2\epsilon^4))$ samples per round for a total of $\tilde{O}(1/(\gamma^3\epsilon^5))$ samples. As each iteration runs in polynomial time, the total running time follows.

When the while loop terminates, we have that $\Pr_{\mathbf{x}\sim\mathcal{D}_\mathbf{x}}[\mathbf{x} \in S^{(i)}] \leq 4\epsilon/3$, i.e., we will have accounted for at least a $(1 - 4\epsilon/3)$-fraction of the total probability mass. Since our algorithm achieves misclassification error at most $\eta + 4\epsilon/3$ in all the regions we accounted for, its total misclassification error is at most $\eta + 8\epsilon/3$. Rescaling $\epsilon$ by a constant factor gives Theorem 2.2. $\qquad\square$

**Remark 2.6.** *If the value of* $\mathrm{OPT}$ *is smaller than* $\eta - \xi$ *for some value* $\xi > 0$, *Algorithm 1 gets misclassification error less than* $\eta - \Omega(\gamma^2\xi^2)$ *when run for* $\epsilon = O(\gamma^2\xi^2)$. *This is because, in the first iteration,* $L^{(1)}(\mathbf{w}^{(1)}) \leq -\gamma(\lambda - \mathrm{OPT})/2 \leq -\gamma\xi/2$, *which implies, by Lemma 2.5, that the obtained error in* $S^{(1)}$ *is at most* $\lambda - \gamma\xi/4$. *The misclassification error in the remaining regions is at most* $\lambda + \epsilon$, *and region* $S^{(1)}$ *has probability mass at least* $\gamma\xi/4$. *Thus, the total misclassification error is at most* $\lambda + \epsilon - \gamma^2\xi^2/16 = \eta - \Omega(\gamma^2\xi^2)$, *when run for* $\epsilon = O(\gamma^2\xi^2)$.

## 2.2 The General Case

In the general case, we assume that $\mathcal{D}_\mathbf{x}$ is an arbitrary distribution supported on $b$-bit integers. While such a distribution might have exponentially small margin in the dimension $d$ (or even $0$), we will preprocess the distribution to ensure a margin condition by removing outliers.

We will require the following notion of an outlier:

**Definition 2.7** ([DV04a]). We call a point $\mathbf{x}$ in the support of a distribution $\mathcal{D}_\mathbf{x}$ a $\beta$-outlier, if there exists a vector $\mathbf{w} \in \mathbb{R}^d$ such that $\langle \mathbf{w}, \mathbf{x}\rangle^2 \leq \beta \, \mathbf{E}_{\mathbf{x}\sim\mathcal{D}_\mathbf{x}}[\langle \mathbf{w}, \mathbf{x}\rangle^2]$.

We will use Theorem 3 of [DV04a], which shows that any distribution supported on $b$-bit integers can be efficiently preprocessed using samples so that no large outliers exist.

**Lemma 2.8** (Rephrasing of Theorem 3 of [DV04a]). *Using $m = \tilde{O}(d^2 b)$ samples from $\mathcal{D}_{\mathbf{x}}$, one can identify with high probability an ellipsoid $E$ such that $\mathbf{Pr}_{\mathbf{x} \sim \mathcal{D}_{\mathbf{x}}}[\mathbf{x} \in E] \geq \frac{1}{2}$ and $\mathcal{D}_{\mathbf{x}}|_E$ has no $\Gamma^{-1} = \tilde{O}(db)$-outliers.*

Given this lemma, we can adapt Algorithm 1 for the large margin case to work in general. The pseudo-code is given in Algorithm 2. It similarly returns a decision list $[(\mathbf{w}^{(1)}, T^{(1)}, E^{(1)}), (\mathbf{w}^{(2)}, T^{(2)}, E^{(2)}), \cdots]$ as output.

---

**Algorithm 2** Main Algorithm (general case)

---

1: Set $S^{(1)} = \mathbb{R}^d$, $\lambda = \eta + \epsilon$, $\Gamma^{-1} = \tilde{O}(db)$, $m = \tilde{O}(\frac{1}{\Gamma^2 \epsilon^4})$.
2: Set $i \leftarrow 1$.
3: Draw $O\left((1/\epsilon^2)\log(1/(\epsilon\Gamma))\right)$ samples from $\mathcal{D}_{\mathbf{x}}$ to form an empirical distribution $\tilde{\mathcal{D}}_{\mathbf{x}}$.
4: **while $\mathbf{Pr}_{\mathbf{x} \sim \tilde{\mathcal{D}}_{\mathbf{x}}}\left[\mathbf{x} \in S^{(i)}\right] \geq \epsilon$ do**
5:     Run the algorithm of Lemma 2.8 to remove $\Gamma^{-1}$-outliers from the distribution $\mathcal{D}_{S^{(i)}}$ by filtering points outside the ellipsoid $E^{(i)}$.
6:     Let $\Sigma^{(i)} = \mathbf{E}_{(\mathbf{x},y) \sim \mathcal{D}^{(i)}|_{S^{(i)}}}[\mathbf{x}\mathbf{x}^T]$ and set $\mathcal{D}^{(i)} = \Gamma \Sigma^{(i)-1/2} \cdot \mathcal{D}|_{S^{(i)} \cap E^{(i)}}$ be the distribution $\mathcal{D}|_{S^{(i)} \cap E^{(i)}}$ brought in isotropic position and rescaled by $\Gamma$ so that all vectors have $\ell_2$-norm at most 1.
7:     Let $L^{(i)}(\mathbf{w}) = \mathbf{E}_{(\mathbf{x},y) \sim \mathcal{D}^{(i)}}[\text{LeakyRelu}_{\lambda}(-y\langle \mathbf{w}, \mathbf{x} \rangle)]$
8:     Run SGD on $L^{(i)}(\mathbf{w})$ for $\tilde{O}(1/(\Gamma^2 \epsilon^2))$ iterations, to get $\mathbf{w}^{(i)}$ with $\|\mathbf{w}^{(i)}\|_2 = 1$ such that $L^{(i)}(\mathbf{w}^{(i)}) \leq \min_{\mathbf{w}:\|\mathbf{w}\|_2 \leq 1} L^{(i)}(\mathbf{w}) + \Gamma\epsilon/2$.
9:     Draw $m$ samples from $\mathcal{D}^{(i)}$ to form an empirical distribution $\mathcal{D}_m^{(i)}$.
10:    Find a threshold $T^{(i)}$ such that $\mathbf{Pr}_{(\mathbf{x},y) \sim \mathcal{D}_m^{(i)}}[|\langle \mathbf{w}^{(i)}, \mathbf{x} \rangle| \geq T^{(i)}] \geq \Gamma\epsilon$ and the empirical misclassification error, $\mathbf{Pr}_{(\mathbf{x},y) \sim \mathcal{D}_m^{(i)}}[h_{\mathbf{w}}(\mathbf{x}) \neq y \,|\, |\langle \mathbf{w}^{(i)}, \mathbf{x} \rangle| \geq T^{(i)}]$, is minimized.
11:    Revert the linear transformation by setting $\mathbf{w}^{(i)} \leftarrow \Gamma \Sigma^{(i)-1/2} \cdot \mathbf{w}^{(i)}$.
12:    Update the unclassified region $S^{(i+1)} \leftarrow S^{(i)} \setminus \{\mathbf{x} : \mathbf{x} \in E^{(i)} \wedge |\langle \mathbf{w}^{(i)}, \mathbf{x} \rangle| \geq T^{(i)}\}$ and set $i \leftarrow i + 1$.
13: Return the classifier $[(\mathbf{w}^{(1)}, T^{(1)}, E^{(1)}), (\mathbf{w}^{(2)}, T^{(2)}, E^{(2)}), \cdots]$

---

Our main result is the following theorem:

**Theorem 2.9.** *Let $\mathcal{D}$ be a distribution over $(d+1)$-dimensional labeled examples with bit-complexity $b$, generated by an unknown halfspace corrupted by Massart noise at rate $\eta < 1/2$. Algorithm 2 uses $\tilde{O}(d^3 b^3/\epsilon^5)$ samples, runs in $\text{poly}(d, 1/\epsilon, b)$ time, and returns, with probability $2/3$, a classifier $h$ with misclassification error $\text{err}_{0-1}^{\mathcal{D}}(h) \leq \eta + \epsilon$.*

# 3 Conclusions

The main contribution of this paper is the first non-trivial learning algorithm for the class of halfspaces (or even disjunctions) in the distribution-free PAC model with Massart noise. Our algorithm achieves misclassification error $\eta + \epsilon$ in time $\text{poly}(d, 1/\epsilon)$, where $\eta < 1/2$ is an upper bound on the Massart noise rate. The most obvious open problem is whether this error guarantee can be improved to $f(\text{OPT}) + \epsilon$ (for some function $f : \mathbb{R} \to \mathbb{R}$ such that $\lim_{x \to 0} f(x) = 0$) or, ideally, to $\text{OPT} + \epsilon$. It follows from our lower bound constructions that such an improvement would require new algorithmic ideas. It is a plausible conjecture that obtaining better error guarantees is computationally intractable. This is left as an interesting open problem for future work. Another open question is whether there is an efficient *proper* learner matching the error guarantees of our algorithm. We believe that this is possible, building on the ideas in [DV04b], but we did not pursue this direction. More broadly, what other concept classes admit non-trivial algorithms in the Massart noise model? Can one establish non-trivial reductions between the Massart noise model and the agnostic model? And are there other natural semi-random input models that allow for efficient PAC learning algorithms in the distribution-free setting?

**Acknowledgments**

Part of this work was performed while Ilias Diakonikolas was at the Simons Institute for the Theory of Computing during the program on Foundations of Data Science. Ilias Diakonikolas is supported by Supported by NSF Award CCF-1652862 (CAREER) and a Sloan Research Fellowship. This research was performed while Themis Gouleakis was a postdoctoral researcher at USC.

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
