[Supplementary Material]

# Supplementary Material

## A  Omitted Proofs from Section 2

We now analyze Algorithm 2 and establish Theorem 2.9. To do this, we need to adapt Lemma 2.3 to the case without margin. We replace the margin condition by requiring that the minimum eigenvalue of the covariance matrix is at least $\Gamma$.

**Lemma A.1.** *Let $\mathcal{D}_{\mathbf{x}}$ be any distribution over points with $\ell_2$-norm bounded by 1, with covariance having minimum eigenvalue at least $\Gamma$. If $\lambda \geq \eta$, then $\min_{\mathbf{w}:\|\mathbf{w}\|_2 \leq 1} L(\mathbf{w}) \leq -\Gamma(\lambda - \eta)$.*

*Proof.* We will show the statement for the optimal unit vector $\mathbf{w}^*$. For any fixed $\mathbf{x}$, we have that

$$\ell(\mathbf{w}^*, \mathbf{x}) = (\mathrm{err}(\mathbf{w}^*, \mathbf{x}) - \lambda)|\langle \mathbf{w}^*, \mathbf{x}\rangle| = (\eta(\mathbf{x}) - \lambda)|\langle \mathbf{w}^*, \mathbf{x}\rangle| \leq -(\lambda - \eta)|\langle \mathbf{w}^*, \mathbf{x}\rangle|.$$

Taking expectation over $\mathbf{x}$ drawn from $\mathcal{D}_{\mathbf{x}}$, we get the statement as

$$\mathbf{E}[|\langle \mathbf{w}^*, \mathbf{x}\rangle|] \geq \mathbf{E}[|\langle \mathbf{w}^*, \mathbf{x}\rangle|^2] \geq \Gamma,$$

where we used the fact that for all points $\mathbf{x}$, $|\langle \mathbf{w}^*, \mathbf{x}\rangle| \leq \|\mathbf{x}\|_2^2 \leq 1$. $\qquad\square$

With Lemma A.1 in hand, we are ready to prove Theorem 2.9. We will use Lemma 2.4 and Lemma 2.5 whose statements do not require that the distribution of points has large margin.

*Proof of Theorem 2.9.* We again consider the steps of Algorithm 2 in every iteration $i$. At every iteration, we consider a distribution $\mathcal{D}^{(i)}$ consisting only of points not handled in previous iterations.

For every iteration, the distribution $\mathcal{D}^{(i)}$ is rescaled so that the norm of all points is bounded by 1 and the covariance matrix has minimum eigenvalue $\Gamma$. Lemma A.1 implies that the minimizer of the loss $L^{(i)}$ has value less than $-\Gamma(\lambda - \eta) \leq -\Gamma\epsilon$.

By the guarantees of Lemma 2.4, running SGD in line 8 on $L^{(i)}(\cdot)$ with projection to the $\ell_2$ ball for $\tilde{O}\left(\frac{1}{\Gamma^2\epsilon^2}\right)$ steps, we can find a $\mathbf{w}^{(i)}$ such that $L^{(i)}(\mathbf{w}^{(i)}) \leq -\Gamma\epsilon/2$ and $\left\|\mathbf{w}^{(i)}\right\|_2 = 1$. This step requires $\tilde{O}\left(\frac{1}{\Gamma^2\epsilon^2}\right)$ samples from $\mathcal{D}^{(i)}$ which translate to at most $\tilde{O}\left(\frac{1}{\Gamma^2\epsilon^3}\right)$ samples from $\mathcal{D}$, as $\mathbf{Pr}_{\mathbf{x}\sim\mathcal{D}_{\mathbf{x}}}\left[\mathbf{x} \in S^{(i)}\right] = \Omega(\epsilon)$.

Then, similar to the proof of Theorem 2.2, Lemma 2.5 implies that there exists a threshold $T \geq 0$, such that:

- $\mathbf{Pr}_{(\mathbf{x},y)\sim\mathcal{D}^{(i)}}[|\langle \mathbf{w}, \mathbf{x}\rangle| \geq T] \geq \Gamma\epsilon$, and

- $\mathbf{Pr}_{(\mathbf{x},y)\sim\mathcal{D}^{(i)}}[h_{\mathbf{w}}(\mathbf{x}) \neq y \,|\, |\langle \mathbf{w}, \mathbf{x}\rangle| \geq T] \leq \eta + \epsilon$.

Line 10 of Algorithm 2 estimates the threshold using samples and $m = \tilde{O}(\frac{1}{\Gamma^2\epsilon^4})$ samples suffice to estimate the probability mass of the region within $\Gamma\epsilon^2$ and the misclassification error within $\epsilon$. This is satisfied for all iterations with constant probability. Note also that the sample complexity for the estimation of $\mathcal{D}_{\mathbf{x}}$ is less than $m$.

The number of iterations required is only $O(\frac{\log(1/\epsilon)}{\Gamma\epsilon})$, as then we will have accounted for $1 - \Theta(\epsilon)$ of the total probability mass. As we get less than $\eta + \epsilon$ misclassification error in all the regions we accounted for, the total misclassification error is at most $\eta + 2\epsilon$.

Thus overall, with constant success probability, Algorithm 2 runs for $O(\frac{\log(1/\epsilon)}{\Gamma\epsilon})$ iterations and draws $\tilde{O}(\frac{1}{\Gamma^2\epsilon^4})$ samples per round for a total of $\tilde{O}(\frac{1}{\Gamma^3\epsilon^5})$ samples. As the algorithm runs in polynomial time per iteration, the proof of Theorem 2.9 is complete. $\qquad\square$

# B Lower Bounds Against Natural Approaches

In this section, we show that certain natural approaches for learning halfspaces with Massart noise inherently fail, even in the large margin case.

We begin in Section B.1 by showing that the common approach of using a convex surrogate function for the 0-1 loss cannot lead to non-trivial misclassification error. (We remark that this comes in sharp contrast with the problem of learning large margin halfspaces with RCN, where a convex surrogate works, see, e.g., Theorem C.1 in Section C).

In Section B.2, we provide evidence that improving the misclassification guarantee of $\eta + \epsilon$ achieved by our algorithm requires a genuinely different approach. In particular, we show that the approach of iteratively using *any* convex proxy followed by thresholding gets stuck at error $\Omega(\eta) + \epsilon$, even in the large margin case.

## B.1 Lower Bounds Against Minimizing a Convex Surrogate Function

One of the most common approaches in machine learning is to replace the 0-1 loss in the ERM by an appropriate convex surrogate and solve the corresponding convex optimization problem. In this section, we show that this approach inherently fails to even give a weak learner in the presence of Massart noise — even under a margin assumption.

In more detail, we construct distributions over a finite sets of points in the two-dimensional unit ball for which the method of minimizing a convex surrogate will always have misclassification error $\min\{1/2, \Theta(\eta/\gamma)\}$, where $\gamma$ is the maximum margin with respect to any hyperplane. Our proof is inspired by an analogous construction in [LS10], which shows that one cannot achieve non-trivial misclassification error for learning halfspaces in the presence of RCN, using certain convex boosting techniques. Our argument is more involved in the sense that we need to distinguish two cases and consider different distributions for each one. Furthermore, by leveraging the additional strength of the Massart noise model, we are able to show that the misclassification error has to be larger than the noise level $\eta$ by a factor of $1/\gamma$.

In particular, our first case corresponds to the situation where the convex surrogate function is such that misclassified points are penalized by a fair amount and therefore the effect of noise of correctly classified points on the gradient is significant. This allows a significant amount of probability mass to be in the region where the true separating hyperplane and the one defined by the minimum of the convex surrogate function disagree. The second case, which is the complement of the first one, uses the fact that the contribution of a correctly classified point on the gradient is not much smaller than that of a misclassified point, again allowing a significant amount of probability mass to be given to the aforementioned disagreement region. Formally, we prove the following:

**Theorem B.1.** *Consider the family of algorithms that produce a classifier* $\mathrm{sign}(\langle \mathbf{w}^*, \mathbf{x} \rangle)$*, where* $\mathbf{w}^*$ *is the minimum of the function* $G(\mathbf{w}) = \mathbf{E}_{(\mathbf{x},y)\sim\mathcal{D}}[\phi(y\langle \mathbf{w}, \mathbf{x}\rangle)]$*. For any decreasing convex* [1] *function* $\phi : \mathbb{R} \to \mathbb{R}$*, there exists a distribution* $\mathcal{D}$ *over* $\mathbb{B}_2 \times \{\pm 1\}$ *with margin* $\gamma \leq \frac{\sqrt{3}-1}{4}$ *such that the classifier* $\mathrm{sign}(\langle \mathbf{w}^*, \mathbf{x} \rangle)$*, misclassifies a* $\min\{\frac{\eta}{8\gamma}, \frac{1}{2}\}$ *fraction of the points.*

*Proof.* We consider algorithms that perform ERM with a convex surrogate, i.e., minimize a loss of the form $G(\mathbf{w}) = \mathbf{E}_{(\mathbf{x},y)\sim\mathcal{D}}[\phi(y\langle \mathbf{w}, \mathbf{x}\rangle)]$, for some convex function $\phi : \mathbb{R} \to \mathbb{R}$ for $\|\mathbf{w}\|_2 \leq 1$. We can assume without loss of generality that $\phi$ is differentiable and its derivative is non-decreasing. Even if there is a countable number of points in which it is not, there is a subderivative that we can pick for each of those points such that the derivative is increasing overall, since we have assumed that $\phi$ is convex. Therefore, our argument still goes through even without assuming differentiability.

We start by calculating the gradient of $G$ as a function of the derivative of $\phi$ at the minimum of $G$. Suppose that $\mathbf{v} \in \mathbb{R}^d$ is the minimizer of $G$ subject to $\|\mathbf{w}\|_2 \leq 1$. This requires that either $\nabla G(\mathbf{v})$ is parallel to $\mathbf{v}$, in case the unconstrained minimum lies outside the region $\|\mathbf{w}\|_2 \leq 1$, or $\nabla G(\mathbf{v}) = \mathbf{0}$. Therefore, we have that for every $i > 1$, the following holds:

$$\frac{\partial G}{\partial \mathbf{w}_i}(\mathbf{v}) = \mathbf{E}_{(\mathbf{x},y)\sim\mathcal{D}}[\phi'(y\langle \mathbf{v}, \mathbf{x}\rangle)(y\mathbf{x}_i)] = 0 \ .$$

Our lower bound construction produces a distribution $\mathcal{D}$ over $(\mathbf{x}, y)$ whose $\mathbf{x}$ marginal, $\mathcal{D}_{\mathbf{x}}$, is supported on the 2-dimensional unit ball. We need to consider two complementary cases for the convex function $\phi$. For each case, we will define judiciously chosen distributions, $\mathcal{D}_1, \mathcal{D}_2$ for which the result holds.

**Case I:** There exists $z \in [0, \sqrt{3}/2]$ such that: $|\phi'(z)| < \frac{1}{2} \frac{\eta}{1-\eta} |\phi'(-z)|$.

In this case, we consider the distribution shown in Figure 1 (left), where the point $(z, -\gamma)$ has probability mass $p$ and the remaining $1 - p$ mass in on the point $(z, \sqrt{1 - z^2})$. We need to pick the parameter $p$ so that $\mathbf{v} = \mathbf{e}_1$ is the minimum of $G(\mathbf{w})$.

Note that the misclassification error is $\mathrm{err}_{0-1}^{\mathcal{D}_1}(\mathrm{sign}(\langle \mathbf{v}, \mathbf{x} \rangle)) = p + (1 - p) \cdot \eta$. The condition that $\mathbf{v} = \mathbf{e}_1$ is a minimizer of $G(\mathbf{w})$ is equivalent to $\mathbf{E}_{(\mathbf{x}, y) \sim \mathcal{D}_1}[\phi'(y\langle \mathbf{v}, \mathbf{x} \rangle)(y\mathbf{x}_2)] = 0$. Substituting for our choice of $\mathcal{D}_1$ with noise level $\eta$ on $(z, -\gamma)$ and $0$ on $(z, \sqrt{1 - z^2})$, we get:

$$p \cdot \phi'(-z) \cdot \gamma + (1 - p) \cdot (1 - \eta)\phi'(z) \cdot \sqrt{1 - z^2} + (1 - p) \cdot \eta \cdot \phi'(-z) \cdot (-\sqrt{1 - z^2}) = 0 \ .$$

Equivalently, we have:

$$(1 - p) \cdot \eta \cdot |\phi'(-z)| \cdot \sqrt{1 - z^2} = p \cdot \gamma \cdot |\phi'(-z)| + (1 - p) \cdot (1 - \eta)|\phi'(z)|\sqrt{1 - z^2} \ .$$

Now, suppose that $|\phi'(z)| = (1 - \alpha)\frac{\eta}{1-\eta}|\phi'(-z)|$, for some $\alpha > \frac{1}{2}$. By substituting and simplifying, we get:

$$p \cdot \gamma = \alpha(1 - p)\eta\sqrt{1 - z^2} = (1 - p)\eta\Delta \ ,$$

where $\Delta = \alpha\sqrt{1 - z^2}$, which in turns gives that

$$p = \frac{\eta\Delta}{\gamma + \eta\Delta} \ .$$

Thus, the misclassification error is

$$\mathrm{err}_{0-1}^{\mathcal{D}_1}(\mathrm{sign}(\langle \mathbf{v}, \mathbf{x} \rangle)) = p + (1 - p)\eta = \eta + (1 - \eta)p = \eta + \frac{(1 - \eta)\eta\Delta}{\gamma + \eta\Delta} = \frac{\eta(\gamma + \Delta)}{\gamma + \eta\Delta} \geq \frac{1}{1 + \frac{\gamma}{\eta\Delta}} \ .$$

Note that for margin $\gamma \leq \eta \cdot \Delta$, we have that $\mathrm{err}_{0-1}^{\mathcal{D}_1}(\mathrm{sign}(\langle \mathbf{v}, \mathbf{x} \rangle)) \geq \frac{1}{2}$, and we can achieve error exactly $\frac{1}{2}$ by setting the point $Q_1$ at distance exactly $\eta \cdot \Delta$. On the other hand, when the margin is $\gamma \leq \eta \cdot \Delta$, we have: $\mathrm{err}_{0-1}^{\mathcal{D}_1}(\mathrm{sign}(\langle \mathbf{v}, \mathbf{x} \rangle)) \geq \frac{\eta\Delta}{2\gamma} \geq \frac{\eta}{8\gamma}$. The last inequality comes from the fact that $\Delta = \alpha\sqrt{1 - z^2} \geq 1/4$, since $\alpha \geq 1/2$ and $z \leq \sqrt{3}/2$.

**Case II:** For all $z \in [0, \sqrt{3}/2]$ we have that $|\phi'(z)| \geq \frac{1}{2} \frac{\eta}{1-\eta} |\phi'(-z)|$.

In this case, we consider the distribution shown in Figure 1 (right), where the only points that have non-zero mass are: $(0, -2\gamma)$, which has probability mass $p$, and $(1/2, -r)$, with mass $1 - p$. We need to appropriately select the parameters $p$ and $r$, so that $\mathbf{v}$ is actually the minimizer of the function $G(\mathbf{w})$, and the misclassification error (which is equal to $p$ in this case) is maximized.

Note that $\mathbf{v}$ satisfies $\mathbf{E}_{(\mathbf{x}, y) \sim \mathcal{D}_2}[\phi'(y\langle \mathbf{v}, \mathbf{x} \rangle)(y \cdot \mathbf{x}_2)] = 0$. Substituting for this particular distribution $\mathcal{D}_2$ with noise level $0$ on both points, we get:

$$p \cdot \phi'(0) \cdot (2\gamma) + (1 - p)\phi'(1/2) \cdot (-r) = 0 \ .$$

Since $\phi'$ is monotone, we get:

$$p|\phi'(0)| \cdot (2\gamma) = (1 - p)|\phi'(1/2)| \cdot r \ .$$

By rearranging, we get:

$$p = \frac{|\phi'(1/2)| \cdot r}{|\phi'(1/2)| \cdot r + 2\gamma|\phi'(0)|} \ .$$

By the definition of Case II and the fact that $\phi$ is decreasing and convex, we have that:

$$|\phi'(1/2)| \geq (\eta/2)|\phi'(-1/2)| \geq (\eta/2)|\phi'(0)| \ .$$

Figure 1: Probability distribution for Case I is on the left and for the complementary Case II is on the right.

Therefore, we can get misclassification error:

$$\mathrm{err}_{0-1}^{\mathcal{D}_2}(\mathrm{sign}(\langle \mathbf{v}, \mathbf{x} \rangle)) = p \geq \frac{|\phi'(1/2)| \cdot r}{|\phi'(1/2)| \cdot r + \frac{4\gamma}{\eta}|\phi'(1/2)|} = \frac{1}{1 + \frac{4\gamma}{\eta r}}.$$

We note that $r$ must be chosen within the interval $\left[0, \sqrt{3}/2 - 2\gamma\right]$, so that the $\gamma$-margin requirement is satisfied.

For margin $\frac{\sqrt{3}-1}{4}\gamma \leq \frac{\eta r}{4}$, we get $\mathrm{err}_{0-1}^{\mathcal{D}_2}(\mathrm{sign}(\langle \mathbf{v}, \mathbf{x} \rangle)) > 1/2$, and we can achieve error exactly $1/2$ by moving the probability mass $p$ from $Q_1(0, -2\gamma)$ to $Q_3(0, -\frac{\eta r}{2})$. If $\gamma \geq \frac{\eta r}{4}$, then $\mathrm{err}_{0-1}^{\mathcal{D}_2}(\mathrm{sign}(\langle \mathbf{v}, \mathbf{x} \rangle)) \geq \frac{\eta r}{4\gamma} \geq \frac{\eta r}{8\gamma}$. The last inequality comes from the fact that we can pick $r = 1/2 \leq \sqrt{3}/2 - 2\gamma$. This completes the proof of Theorem B.1. □

Figure 2: Probability Distribution for Modified Case II.

## B.2 Lower Bound Against Convex Surrogate Minimization Plus Thresholding

The lower bound established in the previous subsection does not preclude the possibility that our algorithmic approach in Section 2 giving misclassification error $\approx \eta$ can be improved by replacing the LeakyRelu function by a different convex surrogate. In this section, we prove that using a different convex surrogate in our thresholding approach indeed does not help.

That is, we show that any approach which attempts to obtain an accurate classifier by considering a thresholded region cannot get misclassification error better than $\Omega(\eta)$ within that region, i.e., the bound of our algorithm cannot be improved with this approach. Formally, we prove:

**Theorem B.2.** *Consider the family of algorithms that produce a classifier* $\mathrm{sign}(\langle \mathbf{w}^*, \mathbf{x} \rangle)$, *where* $\mathbf{w}^*$ *is the minimizer of the function* $G(\mathbf{w}) = \mathbf{E}_{(\mathbf{x},y) \sim \mathcal{D}}[\phi(y\langle \mathbf{w}, \mathbf{x} \rangle)]$. *For any decreasing convex function* $\phi : \mathbb{R} \to \mathbb{R}$, *there exists a distribution* $\mathcal{D}$ *over* $\mathbb{B}_2 \times \{\pm 1\}$ *with margin* $\gamma \leq \sqrt{3}/8$ *such that the classifier* $\mathrm{sign}(\langle \mathbf{w}^*, \mathbf{x} \rangle)$ *misclassifies a* $(1 - O(\gamma)) \cdot \Omega(\eta)$ *fraction of the points* $\mathbf{x}$ *that lie in the region* $\{\mathbf{x} : \langle \mathbf{w}, \mathbf{x} \rangle > T\}$ *for any threshold* $T$.

*Proof.* Our proof proceeds along the same lines as the proof of Theorem B.1, but with some crucial modifications. In particular, we argue that Case I above remains unchanged but Case II requires a different construction.

Firstly, we note that the points $Q_1, Q_2$ in Case I are the only points that are assigned non-zero mass by the distribution and they are at equal distance $z$ from the output classifier's hyperplane. Therefore, any set of the form $\mathbb{1}_{\langle \mathbf{v}, \mathbf{x} \rangle > T}$, where $\mathbf{v}$ is the unit vector perpendicular to the hyperplane, will either contain the entire probability mass or $0$ mass. Thus, for all the meaningful choices of the threshold $T$, we get the same misclassification error as with $T = 0$. This means that the example distribution and the analysis for Case I remain unchanged.

However, Case II in the proof of Theorem B.1 requires modification as the points $Q_1, Q_2$ are at different distances from the classifier's hyperplane.

Here we will restrict our attention to the case where the distances of the two points from the classifier's hyperplane are actually equal and get a lower bound nearly matching the upper bound in Section 2. This lower bound applies, due to reasons explained above, to all approaches that use a combination of minimizing a convex surrogate function and thresholding.

**Modified Case II:** We recall that in this case the following assumption on the function $\phi$ holds: For all $z \in [0, \sqrt{3}/2]$ it holds $|\phi'(z)| \geq \frac{1}{2}\frac{\eta}{1-\eta}|\phi'(-z)|$.

The new distribution $\mathcal{D}_2'$ is going to be as shown in Figure 2. That is, we assign mass $p$ on the point $Q_1(1/4, \sqrt{3}/4 + 2\gamma)$ and mass $1 - p$ on the point $Q_2(1/4, \sqrt{3}/4 - 2\gamma)$.

Similarly to the previous section, we use the equation: $\mathbf{E}_{(\mathbf{x},y)\sim\mathcal{D}}[\phi'(y\langle\mathbf{v},\mathbf{x}\rangle)(y\cdot\mathbf{x}_2)] = 0$, that holds for $\mathbf{v}$ being the minimum of $G(w) = \mathbf{E}_{(\mathbf{x},y)\sim D}[\phi(y\langle\mathbf{w},\mathbf{x}\rangle)]$, to get:

$$p \cdot \phi'(-1/4) \cdot \left(\sqrt{3}/4 + 2\gamma\right) + (1 - p) \cdot \phi'(1/4) \cdot \left[-\left(\sqrt{3}/4 - 2\gamma\right)\right] = 0 \, ,$$

or equivalently:

$$p = \frac{|\phi(1/4)| \left(\sqrt{3}/4 - 2\gamma\right)}{|\phi(1/4)| \left(\sqrt{3}/4 - 2\gamma\right) + |\phi(-1/4)| \left(\sqrt{3}/4 + 2\gamma\right)} \geq \frac{\left(\sqrt{3}/4 - 2\gamma\right)}{\left(\sqrt{3}/4 - 2\gamma\right) + \frac{2(1-\eta)}{\eta}\left(\sqrt{3}/4 + 2\gamma\right)}$$

$$\geq \frac{\left(\sqrt{3}/4 - 2\gamma\right)}{\left(\sqrt{3}/4 + 2\gamma\right)} \cdot \frac{1}{1 + \frac{2(1-\eta)}{\eta}}$$

$$\geq \left(1 - 8\gamma\sqrt{3}/3\right) \frac{\eta}{4(1-\eta)} \, .$$

This completes the proof of Theorem B.2. □

## C   Learning Large-Margin Halfspaces with RCN

In this section, we show that the problem of learning $\gamma$-margin halfspaces in the presence of RCN can be formulated as a convex optimization problem that can be efficiently solved with any first-order method. Prior work by Bylander [Byl94] used a variant of the Perceptron algorithm to learn $\gamma$-margin halfspaces with RCN. To the best of our knowledge, the result of this section is not explicit in prior work.

In order to avoid problems that would arise if the distribution $\mathcal{D}$ is degenerate (i.e., it assigns non-zero mass on a lower dimensional subspace), we introduce Gaussian noise to the points of the distribution. That is, we sample points $\mathbf{x} + \mathbf{r}$, where $\mathbf{r} \sim N(\mathbf{0}, c^2\mathbf{I})$ and $c \triangleq \frac{\gamma}{\sqrt{2\log(2/\gamma\epsilon)}}$.

In particular, we will show that solving the following convex optimization problem:

$$\underset{\|\mathbf{w}\|_2 \leq 1}{\text{minimize}} \quad G_\lambda(\mathbf{w}) = \mathbf{E}_{(\mathbf{x},y)\sim\mathcal{D}}\left[\mathbf{E}_{\mathbf{r}\sim N(\mathbf{0},c^2\mathbf{I})}[\text{LeakyRelu}_\lambda(-y\langle\mathbf{w},\mathbf{x}+\mathbf{r}\rangle)]\right] \, , \tag{1}$$

for $\lambda \triangleq \eta + \frac{c\epsilon}{\sqrt{2\pi}} \approx \eta$ suffices to solve this learning problem.

Intuitively, the idea here is that by adding the right amount of noise $\mathbf{r}$, we make sure that: (a) the probability that the true halfspace misclassifies the noisy version of a point $\mathbf{x}$ is negligible, and (b) if a point is misclassified by the current halfspace, then it has, on average, a significant contribution to the objective function. Therefore, any solution with sufficiently small value yields a halfspace misclassifying a small fraction of points.

As in Section 2.1, we choose the parameter $\lambda$ for the $\text{LeakyRelu}$ function such that $G_\lambda(\mathbf{w})$ has a slightly negative minimum. This is done in order to avoid $\mathbf{w} = \mathbf{0}$ being the minimizer of the function $G_\lambda(\mathbf{w})$. The minimizer for the convex region $\|\mathbf{w}\|_2 \leq 1$ will instead lie in the (non-convex) set $\|\mathbf{w}\|_2 = 1$.

We can solve Problem (1) with a standard first-order method through samples using SGD. Formally, we show the following:

**Theorem C.1.** *Let $\mathcal{D}$ be a distribution over $(d+1)$-dimensional labeled examples obtained by an unknown $\gamma$-margin halfspace corrupted with RCN at rate $\eta < 1/2$. An application of SGD on $G_\lambda(\mathbf{w})$ using $\tilde{O}(1/(\epsilon^2\gamma^4))$ samples returns, with probability $2/3$, a halfspace with misclassification error at most $\eta + \epsilon$.*

The rest of this section is devoted to the proof of Theorem C.1.

We consider the contribution to the objective $G_\lambda$ of a single point $\mathbf{x}$, denoted by $G_\lambda(\mathbf{w},\mathbf{x})$. That is, we define $G_\lambda(\mathbf{w},\mathbf{x}) = \mathbf{E}_{y\sim\mathcal{D}_y(\mathbf{x})}[\mathbf{E}_{\mathbf{r}\sim N(0,c^2I)}[\text{LeakyRelu}_\lambda(-y\langle\mathbf{w},\mathbf{x}+\mathbf{r}\rangle)]]$ and write $G_\lambda(\mathbf{w}) = \mathbf{E}_{\mathbf{x}\sim\mathcal{D}_\mathbf{x}}[G_\lambda(\mathbf{w},\mathbf{x})]$.

We start with the following claim:

**Claim C.2.** $G_\lambda(\mathbf{w}, \mathbf{x})$ *can be rewritten as:*

$$(1 - 2\eta) \cdot \mathbf{E}_{\mathbf{r} \sim N(0, c^2 I)} \left[ |\langle \mathbf{w}, \mathbf{x} + \mathbf{r} \rangle| \mathbb{1}_{h_\mathbf{w}(\mathbf{x}+\mathbf{r}) \neq h_{\mathbf{w}^*}(\mathbf{x})} \right] - (\lambda - \eta) \cdot \mathbf{E}_{\mathbf{r} \sim N(0, c^2 I)} \left[ |\langle \mathbf{w}, \mathbf{x} + \mathbf{r} \rangle| \right].$$

The proof of the claim follows similarly to the proof of Claim 2.1 and is omitted.

Given this decomposition, we move on to show that $G_\lambda(\mathbf{w}^*, \mathbf{x})$ is sufficiently negative for any $\mathbf{x}$ and provide a lower bound on $G_\lambda(\mathbf{w}, \mathbf{x})$ for any unit vector $\mathbf{w}$.

**Lemma C.3.** *For any $\mathbf{x}$ such that $|\langle \mathbf{w}^*, \mathbf{x} \rangle| \geq \gamma$, it holds*

$$G_\lambda(\mathbf{w}^*, \mathbf{x}) \leq -(\lambda - \eta)\gamma/2 = -\tilde{\Omega}(\gamma^2 \epsilon).$$

*Proof.* For any $\mathbf{x}$ such that $|\langle \mathbf{w}^*, \mathbf{x} \rangle| \geq \gamma$, we have that

$$\mathbf{E}_{\mathbf{r} \sim N(0, c^2 I)} \left[ |\langle \mathbf{w}^*, \mathbf{x} + \mathbf{r} \rangle| \right] \geq |\langle \mathbf{w}^*, \mathbf{x} + \mathbf{E}_{\mathbf{r} \sim N(0, c^2 I)}[\mathbf{r}] \rangle| \geq \gamma.$$

Thus, it suffices to show that:

$$\mathbf{E}_{\mathbf{r} \sim N(0, c^2 I)} \left[ |\langle \mathbf{w}^*, \mathbf{x} + \mathbf{r} \rangle| \mathbb{1}_{h_{\mathbf{w}^*}(\mathbf{x}+\mathbf{r}) \neq h_{\mathbf{w}^*}(\mathbf{x})} \right] \leq (\lambda - \eta)\gamma/2.$$

We have that

$$\mathbf{E}_{\mathbf{r} \sim N(0, c^2 I)} \left[ |\langle \mathbf{w}^*, \mathbf{x} + \mathbf{r} \rangle| \mathbb{1}_{h_{\mathbf{w}^*}(\mathbf{x}+\mathbf{r}) \neq h_{\mathbf{w}^*}(\mathbf{x})} \right] \leq \mathbf{E}_{r \sim N(0, c^2)} \left[ r \mathbb{1}_{r \geq \gamma} \right] = \frac{c}{\sqrt{2\pi}} \exp(-(\gamma/c)^2/2).$$

The choice of $c$, implies that $\frac{c}{\sqrt{2\pi}} \exp(-(\gamma/c)^2/2) = \frac{c}{\sqrt{2\pi}} \epsilon\gamma/2 = (\lambda - \eta)\gamma/2$. $\square$

**Lemma C.4.** *For any unit vectors $\mathbf{w}, \mathbf{x}$, it holds*

$$G_\lambda(\mathbf{w}, \mathbf{x}) \geq \frac{2c}{\sqrt{2\pi}} \left( (1 - 2\eta) \mathbb{1}_{h_\mathbf{w}(\mathbf{x}) \neq h_{\mathbf{w}^*}(\mathbf{x})} - \epsilon \right).$$

*Proof.* To bound the second term in Claim C.2, we note that for any $\mathbf{x}, \mathbf{w}$, we have that

$$\mathbf{E}_{\mathbf{r} \sim N(0, c^2 I)} \left[ |\langle \mathbf{w}, \mathbf{x} + \mathbf{r} \rangle| \right] \leq 1 + c \leq 2.$$

To bound the first term, note that for any $\mathbf{x}$ such that $h_\mathbf{w}(\mathbf{x}) \neq h_{\mathbf{w}^*}(\mathbf{x})$, it holds

$$\mathbf{E}_{\mathbf{r} \sim N(0, c^2 I)} \left[ |\langle \mathbf{w}, \mathbf{x} + \mathbf{r} \rangle| \mathbb{1}_{h_\mathbf{w}(\mathbf{x}+\mathbf{r}) \neq h_{\mathbf{w}^*}(\mathbf{x})} \right] \geq \mathbf{E}_{r \sim N(0, c^2)} \left[ r \mathbb{1}_{r \geq 0} \right] \geq \frac{2c}{\sqrt{2\pi}}.$$

Combining the above gives Lemma C.4. $\square$

*Proof of Theorem C.1.* Taking expectation in Lemma C.3, we get that $G_\lambda(\mathbf{w}^*) \leq -\tilde{\Omega}(\gamma^2 \epsilon)$. From the guarantees of SGD (Lemma 2.4), running SGD with $\tilde{O}(1/(\epsilon^2 \gamma^4))$ iterations and samples gives a point $\mathbf{w}$ where $G_\lambda(\mathbf{w}) \leq G_\lambda(\mathbf{w}^*) + O(\epsilon\gamma^2) \leq 0$.

Furthermore, taking expectation in Lemma C.4, we obtain that

$$(1 - 2\eta)\mathbf{Pr}_{\mathbf{x} \sim \mathcal{D}_\mathbf{x}}[h_\mathbf{w}(\mathbf{x}) \neq h_{\mathbf{w}^*}(\mathbf{x})] \leq \epsilon. \tag{2}$$

Overall, the misclassification error of $h_\mathbf{w}$ is equal to

$$(1 - \eta)\mathbf{Pr}_{\mathbf{x} \sim \mathcal{D}_\mathbf{x}}[h_\mathbf{w}(\mathbf{x}) \neq h_{\mathbf{w}^*}(\mathbf{x})] + \eta(1 - \mathbf{Pr}_{\mathbf{x} \sim \mathcal{D}_\mathbf{x}}[h_\mathbf{w}(\mathbf{x}) \neq h_{\mathbf{w}^*}(\mathbf{x})]) = \eta + (1 - 2\eta)\mathbf{Pr}_{\mathbf{x} \sim \mathcal{D}_\mathbf{x}}[h_\mathbf{w}(\mathbf{x}) \neq h_{\mathbf{w}^*}(\mathbf{x})].$$

From (2) we obtain that the above is at most $\eta + \epsilon$. This completes the proof of Theorem C.1. $\square$

## Footnotes

[1]The function is not necessarily differentiable. In case it isn't, being *convex* means that the subgradients of the points are monotonically increasing.