[Reviews · NeurIPS 2019]

Reviewer 1



The paper gives a PAC learning algorithm for the basic problem of halfspaces in a model of learning with noise. The algorithm uses ideas from previous related results in the simpler model of random classification noise, with important new ideas. Learning with noise is a basic topic in learning theory. It can be argued that the most studied models (random misclassification noise and malicious noise) are unrealistically benign (even though the related SQ model is very important) or malicious, and there is a great need for the study of more realistic models. The Massart noise model is a candidate for such a model. As positive learnability results in the general PAC model were not known for this kind of noise, the result of the present paper is quite significant. The algorithm is non-proper, with a kind of decision list as hypothesis. This is especially interesting, as this class of decision lists is a natural class, which have already been studied (called neural decision lists, linear decision lists and threshold decision lists, going back to the work of Marchand, Golea and Rujan 30 years ago). It would be useful to comment on the possibility of proper learning halfspaces in this model. Comments: It is mentioned that an equivalent notion called malicious misclassification noise'' was studied (also 30 years ago) by Sloan. An explanation (or at least a reference) should be given for the equivalence of the two definitions. Malicious misclassification noise seems to be an appropriate term fitting the general terminology, and it seems that instead of using two names for the equivalent notions, one should just use malicious misclassification noise, noting that the other one is an equivalent definition. Massart noise is also unjustified as the cited paper is due to Massart and Nedelec. A small additional point is that presumably there is some tameness'' assumption for the noise probability function $\eta(x)$ (or not? does this matter for the equivalence proof?). A related comment: the literature review does not distinguish between the real-valued and Boolean domains (for example, Daniely's negative result already holds for the Boolean case); some comments on that should be added. One more terminological remark: the basic PAC model is distribution-independent, and for PAC learning under a fixed distribution'' is used when the underlying distribution is fixed. Thus the term distribution-independent'' in the title seems redundant (of course it is important to emphasize this feature as opposed to results on tame'' distributions in the text).

Reviewer 2



This paper studies the problem of learning halfspaces under arbitrary data distributions and when label noise is present in the data. This problem has a rich history and the celebrated result of [BFKV'97] showed that there exists a polynomial time learning algorithm when the label noise is i.i.d., i.e., when each label is flipped independently with probability eta < 1/2. Essentially this is the only noise model for which we know distribution independent learning results. At the other extreme we have the agnostic learning model where we know that learning halfspaces under the uniform/log-concave distributions is easy and there is also evidence that agnostic learning under arbitrary distributions is hard. An intermediate noise model is the Massart/bounded noise model, where the label of each example x is flipped independently with probability p_x < eta < 1/2. Even for this model it has been a longstanding open problem as to whether one can design a learning algorithm that for any eps>0, achieves error OPT + eps, where OPT is the error of the best halfspace. Before this paper it was known how to achieve this only for uniform/log-concave distributions. The main result of the paper is that for learning halfspaces under arbitrary distributions under Massart noise, one can achieve error eta+ eps in polynomial time. This is a significant advance over the state of the art. The paper shows that one can construct a decision list of halfspaces to achieve this bound. The main insight in achieving this is Lemma 2.5 that states that under Massart noise, if data distribuition has some non-trivial margin, then by minimizing a convex proxy one will end up with a halfspce w that does well on a non-trivial amount of the data distribution. Furthermore, this space can be identified by simply thresholding |w.x| at a certain value T. Once this is proved, one immediately obtains learning algorithm for large margin distributions by simply repeating the process on the distribution that does not fall within the threshold. For the general case, one can use the idea of [BFKV] to preprocess the data so that a large fraction satisfies good margin and then apply lemma 2.5. I very much enjoyed reading the paper, it makes progress on a long standing open problem and will lead to further theoretical work in the area. This is a very strong submission and I absolutely recommend acceptance.

Reviewer 3



This paper provides an efficient algorithm for distribution-free PAC learning of halfspaces under Massart noise. This resolves a long-standing open question, at least for representations with a bounded bit-complexity. The paper is very well-written; the techniques are interesting and well explained, and the organization is good. In several points, there are sloppy statements which are incorrect as stated, though they all seem fixable. The detailed comments below list these issues. I request that the authors address these issues in their response, as well as fix the final version of this submission. The main observation exploited in this work is that while optimizing a convex surrogate cannot achieve the required error in this case (which the authors also prove), there does exist a convex surrogate that achieves a small error on some non-negligible region of the space. Repeatedly minimizing this convex surrogate on the remaining part of the space obtains a low error on the entire space, using improper learning with a decision list of half-spaces. The paper presents the solution to learning half-spaces with a margin and Massart noise, and later generalizes the solution to half-spaces without a margin, using the finite bit-complexity and an additional preprocessing step. The proofs in the body of the paper seem correct, except for some fixable issues listed below. This paper addresses an important and interesting question, and resolves it with an elegant and well-explained technique. Assuming all the small issues are fixed, I strongly recommend that this paper be accepted. Detailed comments: ~~~~~~~~~~~~~~~~~~ 1. The dependence of the solution for the zero-margin case in the bit-complexity of the representation is not revealed until section 1.1. Since the paper claims to resolve an open problem, it is important to discuss the relationship between the original statement of the open problem and the actual solution. 2. In Alg 1, there seems to be an assumption that the marginal distribution is completely available (line 3), otherwise an estimation process is needed here. Please explain. 3. Page 6, lines 249-252 ignore the fact that the guarantee of lemma 2.4 is on expectation only. This is then addressed in page 7, lines 276-279 but it is too late, as the paragraph on page 6 is incorrect as stated. Please unite these two paragraphs and put these on page 6. 4. Proof of lemma 2.5. page 7: the last display equation is incorrect. I believe there are several typos there. The conclusion in line 261 is correct though. Please fix this and explain your correction. 5. Page 7, lines 276-279: It is proposed to use Markov's inequality. Please comment on the boundedness conditions that allow you to do that. Also, Markov's inequality cannot actually get the same guarantee as that of the expectation, there will be some constant factor. Finally, when the procedure is repeated, the right w(i) needs to be selected by estimating L(w(i)) from samples. Please address this. 6. Page 7, line 280: (i) seems to require a small enough lambda. 7. Alg 1, line 7: unclear what is meant by "uniform over the samples". Please rephrase. 8. Page 3, line 101, it is not clear what "on the unit ball" refers to, though I assume this is an assumption on x. Please rephrase. 9. Page 4, line 185: theta is not used.

[Author Response · NeurIPS 2019]

We would like to thank the reviewers for their careful consideration of our paper and their positive feedback. Below we address the comments and questions asked by the reviewers.

**Reviewer #1:** The existence of an efficient proper learner with the same accuracy guarantee is left as an open problem.

The equivalence between Sloan's "malicious misclassification noise" model and the Massart model is well-known in the literature, see, e.g., the introduction of [ABHU15].

By the definition of the Massart model, the *only* assumption on $\eta(x)$ is that $\eta(x) \leq \eta < 1/2$ for all $x$ in the domain.

The Boolean-valued setting is an important special case that captures the difficulty of the problem of learning halfspaces. In particular, Sloan's open problem was explicitly phrased for Boolean disjunctions, a *very* special case of Boolean halfspaces. (This can also be found in Avrim Blum's FOCS 2003 tutorial cited and linked from our paper.)

We chose to emphasize the "distribution-independent" aspect in the title to clarify the distinction with the previous references that learn halfspaces with Massart noise *under the uniform distribution on the unit sphere*, e.g., [ABHU15].

We agree that emphasizing which arguments hold generally and which depend on the Massart noise assumption is useful for providing intuition and we will revise accordingly. The main place where the Massart noise assumption is being used is to show that a vector $\mathbf{w}$ with negative loss $L(\mathbf{w})$ exists (Lemmas 2.3 and A.1). The remaining of the arguments and in particular Lemma 2.5 hold more generally, assuming we can find a vector with negative loss.

**Reviewer #3:** The complexity of our algorithm is polynomial in the dimension $d$, the error $\epsilon$, and the bit complexity $b$ of the examples. The reviewer commented on the dependence on $b$.

In terms of *computational complexity*, a polynomial dependence on $b$ exists in *all known learning algorithms for halfspaces* — even in the realizable (noiseless) case. In fact, it is well-known that removing such a dependence on $b$ in the runtime (even for the realizable case) amounts to developing a *strongly polynomial* time algorithm for general linear programs — a major open problem in theoretical computer science. (This is stated in lines 66-67 of our paper.)

In terms of *sample complexity*, even for the special case of Random Classification Noise all known algorithms — including [BFKV97, Coh97] – have a polynomial dependence on $b$ as well. The reason is that the outlier removal lemma of [BFKV97, DV04a], used as a preprocessing step to create a margin condition, requires this many samples.

*Statement of Open Problem:* As we explain in the introduction of our paper, several authors have posed related versions of the open problem we study. Sloan's original open problem [Slo88, Slo92] asks whether there is an efficient learning algorithm in the Massart noise model for Boolean disjunctions — i.e., OR functions on $\{0,1\}^d$ — *a very special case* of Boolean halfspaces. (Note that $b = d$ when the domain is the Boolean hypercube.) As pointed out in Avrim Blum's FOCS'03 tutorial [Blu03] (lines 48-54 of our paper), and additional personal communication with him, even the *weak learning* version of this problem remained open. Cohen [Coh97] asked whether there is an efficient learning algorithm for halfspaces with Massart noise. For the important setting of Boolean halfspaces, i.e., halfspaces on $\{0,1\}^d$ — already a broad generalization of Sloan's open problem — our algorithm has $\text{poly}(d/\epsilon)$ sample complexity and runtime.

*Estimation in Line 3 of Alg 1:* Indeed, checking the termination condition requires estimating the probability which can be done via sampling. The number of samples required, i.e., $O(\frac{1}{\epsilon^2} \log(1/\gamma\epsilon))$, is much smaller than the number of samples needed in every iteration. We will make a note of that in the revised version of our paper.

*Conversion of SGD guarantees to high probability:* Given that our loss function $L$ is bounded in $[-1, 1]$, we can obtain high probability guarantees of Lemma 2.4 by running SGD multiple times. Here is the simple argument in more detail, which we will include in the revision: At a single run, Markov's inequality for the nonnegative random variable $L(\bar{\mathbf{w}}) - L(\mathbf{w}^*)$ gives:
$$\mathbf{Pr}[L(\bar{\mathbf{w}}) - L(\mathbf{w}^*) \leq 2(\mathbf{E}[L(\bar{\mathbf{w}})] - L(\mathbf{w}^*))] \geq 1/2.$$
From the guarantee that $\mathbf{E}[L(\bar{\mathbf{w}})] \leq L(\mathbf{w}^*) + \epsilon$, this means that with probability at least $\frac{1}{2}$, we can find a vector $\bar{\mathbf{w}}$ with loss at most $L(\mathbf{w}^*) + 2\epsilon$. Running SGD $O(\log(1/\delta))$ times, there exists such a vector with probability at least $1 - \delta$. Identifying such a good vector requires estimating the loss within $\epsilon$ for all the returned vectors, which requires $\tilde{O}(\log(1/\delta)/\epsilon^2)$ samples in total for all vectors, as the loss is bounded in $[-1, 1]$. Thus, the total sample complexity is at most $\tilde{O}(\log(1/\delta)/\epsilon^2)$ to get the guarantee with probability at least $1 - \delta$.

*Typo at the end of Lemma 2.5:* Indeed there is a typo in the last displayed equation, which we will rephrase to make the proof cleaner. The integral from $\bar{T}$ to 1 is lower-bounded by $-\lambda\mathbf{Pr}[|\langle\mathbf{w}, \mathbf{x}\rangle| \geq \bar{T}]$ (second to last displayed equation) and is also upper-bounded by $L(\mathbf{w})/2$, as the integral from 0 to $\bar{T}$ is non-negative. This directly yields that $\mathbf{Pr}[|\langle\mathbf{w}, \mathbf{x}\rangle| \geq \bar{T}] \geq |L(\mathbf{w})|/2\lambda$, as required to complete the proof.

*Comment 6:* We only need that $\lambda$ is sufficiently close to $\eta$ to get a meaningful bound. For large values of $\lambda$, the statement still trivially holds (but is vacuous). *Comments 7 and 8:* We will rephrase for clarity. *Comment 9:* Theta will be removed.

[Meta-Review · NeurIPS 2019]

This is a very strong paper that makes impressive progress on the long-standing open problem of efficiently PAC learning halfspaces under the Massart noise model. While resolving the problem would involve getting within epsilon of the optimal error, achieving eta + epsilon is a breakthrough and likely will fuel future results in learning theory.